# Object-Centric Generative Models for Spatial Scene Understanding

## Abstract

Representing a scene and its constituent objects from raw sensory data is a core ability for enabling robots to interact with their environment. In this paper, we propose a novel system for scene understanding, leveraging object-centric generative models. We demonstrate an agent that is able to learn and reason about 3D objects in an unsupervised fashion and is able to infer object category and pose in an allocentric reference frame. Our agent can infer actions to reach a given, object-relative target viewpoint in simulation, outperforming a supervised baseline trained on the same object set.

## 1 Introduction

Spatial scene understanding is a core ability for enabling robots to understand and interact with their environment, and has been a long standing challenge in computer vision. Humans naturally decompose scenes into object-centric representations and infer information about objects, their appearance and constituent parts, as well as their pose and shape in the 3D space (Hinton, 1979). For example, when seeing a coffee cup, humans know immediately where to reach for the handle, even when the handle is not directly in view.

In the past decade, advances in deep learning have enabled to devise systems that can distinguish objects in 2D images, i.e. by segmentation (Minaee et al., 2021), and/or can predict the object pose in 3D (Xiang et al., 2018a; Du et al., 2021) by training supervised on a dataset of predefined object classes. Also unsupervised methods have been proposed to infer separate objects from a single view (Eslami et al., 2016; Bear et al., 2020). In order to have full 3D scene understanding, other approaches learn representations of complete 3D shapes by training on 3D CAD models of objects of interest (Wu et al., 2015). Such representations can then be used to infer objects shape and pose in 3D from one or more RGB-D images (Sucar et al., 2020).

In contrast, humans learn by actively engaging and interacting with the world (James et al., 2014). A prevailing account of human perception is that the brain builds a generative model of the world, constantly explaining its observations (Friston et al., 2017). In this regard, vision is cast as inverting a generative model of the scene, in order to infer its constituent objects, their appearance and pose (Parr et al., 2021). This is consistent with findings in the brain, where visual inputs are processed by a dorsal ("where") stream on the one hand, representing where an object is in the space, and a ventral ("what") stream on the other hand, representing object identity (Mishkin et al., 1983). In similar vein, Hawkins et al. (2017) hypothesize that cortical columns in the neocortex build object-centric models, capturing object identity and pose in a local reference frame, encoded by cortical grid cells.

In this paper, we propose a novel method for spatial scene understanding, using an agent that can actively move the camera in the scene. For each (novel) object category encountered, an object-centric generative model is trained unsupervised, by training using next view prediction. The agent maintains an ensemble of such object-centric models, that then cast "votes" for the presence of particular objects in the scene. By aggregating these votes over time, as well as space using the known allocentric reference frame of the camera, a global scene representation in inferred. We test our approach in a simulation environment, in which the agent has to reach a goal viewpoint relative to a particular target object in the scene. We show that our agent first gathers observations to detect the different objects in the scene, and once the target object is found, is able to infer the action towards the target viewpoint. We evaluate our approach both qualitatively by visualizing the

emerging behaviour, as well as quantitatively by comparing the final viewpoint to a supervised pose estimation approach.

## 2 MODULAR, OBJECT-CENTRIC MODELS

We propose a system that is able to estimate object identity, position and pose from observations in an enactive agent scenario. The artificial agent "lives" in a simulated environment in which objects are put on a tabletop in different poses. It can freely move a camera around by issuing actions, i.e. relative camera transformations with respect to the current camera pose. One could think of a robotic arm with a wrist-mounted camera to embody such an agent. The goal of the agent is to infer an accurate representation of the workspace. To do so, the agent learns object-centric representations of each encountered object (category). These representations offer the agent an implicit 3D understanding of the object, by predicting "how will this look like" from another viewpoint.

To learn such object-centric representations we use cortical column networks (CCN) (Van de Maele et al., 2021). A CCN is a modular, brain-inspired representation that separates a "what" and a "where" latent code. For each object category, a specialized CCN is trained in an unsupervised fashion, which allows the system to scale to more categories as they are encountered. Figure 1 shows the information flow of a CCN. The encoder processes an image patch where the object of interest is positioned in the center, and then infers the distribution over the pose latent $z_{pose}$, and identity latent $z_{id}$. The transition model infers the distribution over the pose latent $z_{pose}$ after executing an action $a_t$. Finally, the latent codes can be decoded into an expected observation $\hat{o}$ using the decoder.

When an unknown object category is encountered, the agent collects data by looking at the object from different viewpoints and taking center crops, effectively sampling $(o_t, a_t, o_{t+1})$ triplets. We model $z_{pose}$ as an isotropic Gaussian and the identity latent $z_{id}$ as a Bernoulli variable, which indicates whether the image patch belongs to the modeled object category. The model is trained end-to-end as a variational autoencoder (Rezende et al., 2014; Kingma & Welling, 2014), and $z_{id}$ is further regularized by sampling random other patches as negative anchors. After training, a CCN can detect whether an image patch "belongs" to the trained object category, by evaluating the identity latent $z_{id}$, as well as assessing the log evidence of the decoder, which we call a "vote". Note that the pose representation is an implicit pose encoding, relative to the object.

While each CCN model can only estimate the identity of the object category on which the respective CCN was trained, the "votes" from each model can be aggregated into a categorical distribution for each timestep and can be aggregated over time using a Dirichlet process. Crucially, the ensemble of CCNs can detect whether novel object categories are detected, when none of the CCNs finds enough evidence over time. This way, a novel CCN can be instantiated and trained, in a continual learning fashion, when novel object categories are found.

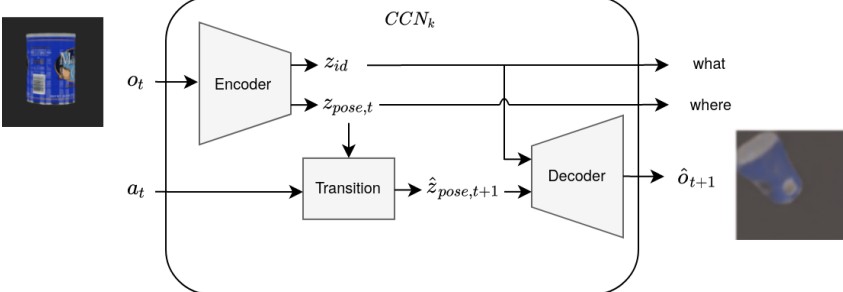

Figure 1: Each CCN model is comprised of an Encoder, Transition and Decoder block which are parameterized by deep neural networks. The encoder infers two latent codes, $z_{id}$ representing the object identity, $z_{pose,t}$ encoding the camera pose in a learnt, object-local reference frame. The Transition model predicts the next (latent) pose given the current pose and action. The Decoder reconstructs a view given the pose.

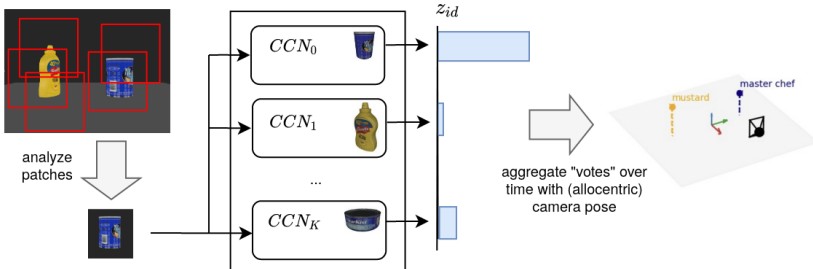

Figure 2: To understand a scene, new observations are split in different image patches, which are fed to all CCN models, each trained on a separate object category. For each patch, each CCN can cast a "vote" based on the evidence that this patch belongs to a certain object class. These votes are aggregated with the camera pose in an allocentric reference frame to infer the different objects, their identity and pose.

To understand a whole scene, the agent considers full resolution camera observations of multiple objects arranged on the table. Now, we feed image patches to the ensemble of CCNs, which will act as pattern detectors for distinct objects. If a CCN votes for a certain image patch, this provides evidence for a particular object being present at that location. As the agent knows its camera pose in the allocentric reference frame, it can estimate the $x, y, z$ position of the object by back-projecting the camera ray centered at the image patch considered. By clustering and aggregating votes for each object candidate over time, the agent converges to a scene representation comprising a list of objects, their object category and allocentric position. This workflow is visualized in Figure 2. In case the camera also provides depth information, the depth points of the identified patches can be exploited to better estimate the object position. We can also exploit the depth information to cut away the ground plane in preprocessing.

In order to evaluate scene and object understanding, we provide the agent with a particular observation that it should reach. Intuitively, this is similar to the case of reaching for a cup's handle, even when not in view. Concretely, we start from a side view on the workspace, and query a particular top view of one of the objects, as depicted on Figure 3. To reach the target view, the agent adopts an active inference scheme to infer actions, which balances actions that on the one hand provide information for the model, and on the other hand bring the agent closer to the goal (Van de Maele et al., 2021). In practice, we first infer the target object id and pose (i.e. $z_{pose,target}$ and $z_{id,target}$) by encoding the target observation using the CCNs. Next, we search for a cluster in the scene representation that matches $z_{id,target}$, and point the camera towards the estimated center. Now, the agent can exploit the CCN transition model to find actions that drive the pose towards $z_{pose,target}$ using Monte Carlo sampling. The agent then takes a step in this direction and the process repeats.

## 3 EXPERIMENTS

To evaluate our approach, we built a simulation environment where objects of the YCB dataset (Calli et al., 2015) are spawned at distinct positions on a tabletop. To train CCNs, the agent is provided with each distinct object, and gathers a dataset by randomly looking at the object from various viewpoints. For parameterizing and training the models, we follow (Van de Maele et al., 2021).

Figure 3 shows a qualitative evaluation of the system, when provided with target top views of both objects. We plot the estimated allocentric object coordinates inferred by the system, as well as the camera trajectory followed to reach the target pose. In both cases, the agent correctly identifies the object identities, and is able to quickly infer the correct pose, exploiting the implicit 3D understanding of the CCN.

We also evaluate quantitatively by comparing with the PoseCNN model for object pose estimation (Xiang et al., 2018b). In this case, PoseCNN is trained supervised to infer the 6-DOF pose of YCB objects. The target viewpoint is found by estimating both the pose of the target and initial observation, and applying this relative transform to the extrinsic camera matrix. Table 1 shows the results in terms of translation and rotation error compared to the ground truth target pose, as well as

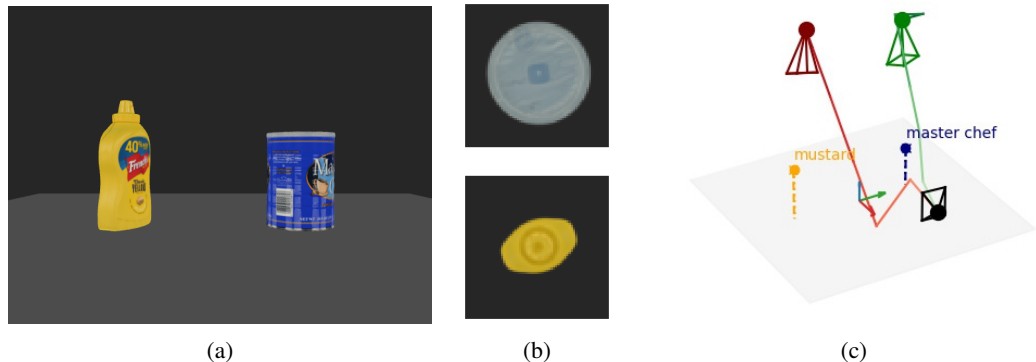

|       (a)       |       (b)       |       (c)       |

Figure 3: (a) Observation in the simulated environment, containing the chips can and the master chef can from the YCB dataset. (b) Preference observations for master chef can (top) and chips can (bottom). (c) Trajectory generated by an agent, driving to a target observation. The ground truth object centers are marked by a cross, while the estimated object center is marked by a dot. The dashed line projects the object to the ground surface.

mean squared error in pixel space compared to the target observation, averaged over 20 randomly generated scenes. Our approach outperforms PoseCNN in both translation error and mean squared error.

## 4 CONCLUSION

In this paper, we presented a novel approach on scene understanding using modular, object-centric generative models. Our system learns a separate generative model for each object category, dubbed CCN, that implicitly encodes object identity and pose in a latent representation. By aggregating evidence from an ensemble of CCNs, and combining this with pose information of the camera, the agent can build a scene representation inferring the different objects, their identity and position in an allocentric reference frame. Moreover, the agent can exploit the implicit pose encoding to infer actions that bring it to a target viewpoint. We showed that our method outperforms a supervised method for pose estimation for matching the target viewpoint, both in translation and pixel error. Also, thanks to the modular CCN architecture, the agent can detect novel object categories, and train additional CCNs, without catastrophic forgetting.

Despite these promising results, there are a number of important issues that need to be addressed in future work. First and foremost, an important question is to what extent this architecture can scale to a large number of object categories. Another current limitation is that now each CCN is trained completely separately, and an object category is only represented by a single, unique instance. An interesting approach would be to share parts of the parameters between CCNs, i.e. the initial layers of the encoder and/or the final decoder layers. Also, instead of encoding the identity in a single Bernoulli variable, one could opt for a more expressive isotropic Gaussian latent that can encode the object appearance. Deciding which object instance should be modeled by which CCN (or instantiate a novel CCN) then becomes an apparent research question. In addition, as our current CCN models are trained on unoccluded object observations, it underperforms in cluttered environments with lots of occlusions. Training on randomly perturbed or cluttered observations might mitigate this and make the CCN model more robust to such perturbances. Nevertheless, we believe that object-centric representations in combination with enactive agents will prove crucial for scene understanding and reasoning.

|  | Translation error | Rotation error | Mean squared error |
|---|---|---|---|
| PoseCNN | 0.361 ±0.138 | 1.986 ±0.714 | 0.061 ±0.023 |
| **Ours** | **0.099** ±0.157 | 1.598 ±0.972 | **0.043** ±0.034 |

Table 1: Error of target camera pose and observation.

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
