# OpenReview forum: "Object-centric Generative Models for Spatial Scene Understanding"
_ICLR.cc/2022/Workshop/OSC — Submitted to ICLR2022 OSC _

### Official Review · Reviewer_6FXG · 2022-03-14
**Object-centric Generative Models for Spatial Scene Understanding**

**Rating:** 2
**Confidence:** 2

**Review:**

Summary:
This work builds on Van de Maele et al. [1] who introduce Cortical Column Networks [CCNs]. A CCN encodes an image patch containing an object into an object ID latent variable and a pose latent variable. The pose latent variable can be combined with an action to compute a transition which allows an object to be decoded with reference to a different viewpoint. An ensemble of CCNs is trained whereby each model in the ensemble is trained to encode one type of object. The CCNs are trained using a simulated dataset that consists of images of individual objects from different viewpoints. Random patches are used as negative examples for each object type. For a scene containing one or multiple objects, each CCN casts a vote towards the identity of an object. Votes are aggregated over space and time. Given an observation and a reference image, the authors use the CCNs to find the viewpoint of the reference image. Quantitatively, this performs favourably compared to PoseCNN. After taking a look at [1], my understanding is that the main extensions of this work are the consideration of multi-object scenes as well as the quantitative comparison to PoseCNN.

Clarity:
The paper reads reasonably well, though a clear statement of how this work relates to and differs from [1], which is cited in the paper, is missing and should be added.

Significance / originality / novelty:
CCNs as introduced in [1] are fairly different to other object-centric models for scene understanding and it is interesting to have some more exploration and investigation of this approach. The paper is a rather incremental extension of [1], but the additional aspects introduced here (application to multi-object scenes and comparison to PoseCNN) are interesting.

Strengths:
- The paper is fairly clear.
- The paper is relevant to the workshop.
- The paper explores an interesting approach for object-centric scene representation learning.
- The authors provide some interesting discussion in the conclusion section on the current limitations of this work as well as possible improvements.

Weaknesses:
- The work is a rather incremental extension of [1].
- The quantitative analysis and discussion is quite minimal. It would be interesting if the authors could should some light on *why* CCNs perform better than PoseCNN. Otherwise, it is difficult to take much away from these results.

[1] “Disentangling What and Where for 3D Object-Centric Representations Through Active Inference”, Van de Maele et al., 2021

---

### Official Review · Reviewer_UEo9 · 2022-03-17
**Relevant work but core claims are not presented well enough**

**Rating:** 1
**Confidence:** 3

**Review:**

The paper describes a system which trains single-object networks to estimate object identity/presence, pose, and action-conditioned transitions. Each network is trained using single-object images (from a grid of viewpoints) and the associated camera parameters. The networks act as an ensemble to classify objects–a feature which the authors argue enables scalability to new objects.

Unfortunately, I don’t suggest accepting at this stage. While the paper is definitely relevant to the workshop, the case for such a system needs to be made better before it can be discussed in a broad forum. As it stands, the paper is weak on multiple levels (in increasing order of significance):

1. The results are bare.
    - In the qualitative case (Fig 3), a single image with two objects is presented. It is unclear whether the object views in (b) are decoded by the model or simply target views. The generated trajectory in (c) is also confusing: why does the camera take a zig-zag path to get above the mustard?
    - Quantitatively, the errors presented in Table 1 show that CCNs perform better than PoseCNN. But the absolute errors don’t mean anything. It could be that both models are underperforming significantly. What physical units are the translation and rotation expressed in? Consider adding a normalized measure of prediction error such as R-squared.

2. The assumptions and descriptions are unclear.
    - If camera parameters and single-object training data are key ingredients of your system, it would be worth specifying that the model is supervised on that data rather than labeling it “unsupervised.”
    - Are object-centered input patches assumed as an input, or are the bounding boxes inferred? I suspect the former. But Figure 2 suggests that object-centered input patches may be further augmented with empty or random patches. Please clarify.
Likewise, please clarify if you’re using RGB-D images.
    - The voting mechanism is not explained well enough. For instance–do you use a softmax across the predicted single-class logits? How do you trade off between the predicted class probabilities and the decoder log evidence?
3. The claims/insights are not argued well enough.
    - In Section 2, the claim is that training a specialized network per object category can allow “the system to scale to more categories as they are encountered.” This depends on (a) what determines an object category in the first place (is a can of Coca Cola different from a can of Cambell’s soup?), (b) the availability of single-category data from new categories to supervise on, and (c) how the “voting” mechanism scales with the addition of new categories. The ideas in the Conclusion (e.g. to encode object identity using more expressive latents which could help generalize across object appearance) cast further doubt on the claim on scalability.
    - At the end of Section 2, the authors argue that the CCN transition model can be used to drive the pose toward $z_{pose, target}$ using “Monte Carlo sampling.” This seems to trivialize the challenge of planning. Even with a perfect transition model were accessible, Monte Carlo sampling would be inadequate and sample-inefficient in most useful scenarios.

Nit (further suggestions to improve the writing):
1. My initial reading of “target object id and pose” was that the objects might need to be manipulated as a goal. This could be cleared up, because a “target object pose” is not the same as a “target camera pose.”
2. The inspiration from what/where processing in the human brain and “enactive” agents is rather loose. Object-centric pose estimation and planning are established problems in the field. The paper could benefit from strengthening its core claims instead.
3. I don’t think (Hinton, 1979) is the best reference to assert that “humans naturally decompose scenes into object-centric representations and infer information about objects, their appearance and constituent parts, as well as their pose and shape in the 3D space.”
4. Figure 3 caption says “chips can” whereas the images show a mustard can.

---

### Decision · Program_Chairs · 2022-03-21

**Decision:**

Reject

**Comment:**

This paper is relevant to the workshop and introduces an interesting extension of “Disentangling What and Where for 3D Object-Centric Representations Through Active Inference” (Van de Maele et al., 2021). Both reviewers, however, agree that the quantitative analysis is too minimal and somewhat inconclusive. Since there is likely insufficient time until the camera-ready deadline to fully address the reviewers' comments, we recommend submitting a revised version of this work to a later venue.